# Effectiveness of thoracic spine manipulation for upper quadrant musculoskeletal disorders: protocol for a systematic review

Erik Thoomes ,[1,2] Gus Tilborghs,[3] Nicola R Heneghan ,[4] Deborah Falla ,[1] Marloes de Graaf[2,3]

¹School of Sport, Exercise and Rehabilitation Sciences, University of Birmingham College of Life and Environmental Sciences, Birmingham, UK
²Research Department, Fysio-Experts, Hazerswoude, The Netherlands
³Department of Manual Therapy, Breederode University of Applied Science, Rotterdam, The Netherlands
⁴School of Sport, Exercise and Rehabilitation Sciences, University of Birmingham, Birmingham, UK

**Correspondence to**
Erik Thoomes;
ejt979@student.bham.ac.uk

## ABSTRACT

**Introduction** Upper quadrant musculoskeletal disorders (UQMD), comprising of cranial, cervical, shoulder and upper extremity disorders, are among the most frequently reported disorders in clinical practice. Thoracic high velocity low amplitude thrust (Tx-HVLAT) manipulation is a form of conservative management recommended in systematic reviews as an effective treatment option for aspects of UQMD disorders such headache, shoulder pain and lateral elbow pain. However, no recent systematic reviews have assessed the effectiveness across UQMD. Therefore, this systematic review aims to update the current evidence on the effectiveness of Tx-HVLAT for patients with UQMD on (1) patient-reported outcomes, (2) performance measures or (3) psychosocial outcomes.

**Methods and analysis** The Cochrane Controlled Trials Register, MEDLINE, EMBASE, CINAHL, PEDro and Index to Chiropractic Literature will be searched from inception using Medical Subject Headings (MeSH), Thesaurus and/or free-text words. Combinations will be made based on localisation, disorder, intervention and design. Following guidelines as advised by the Cochrane Back Review Group, published randomised controlled trials will be included. Two review authors will independently assess the risk of bias (ROB) using the Cochrane Back Review Group's recommended ROB2 tool and will independently extract the data using a standardised data extraction form. Overall quality of the evidence will be evaluated using the Grading of Recommendations, Assessment, Development and Evaluation (GRADE) method. For continuous data, we will calculate standardised mean differences with 95% CIs. For dichotomous outcomes, relative risks and 95% CIs will be calculated. Where possible we will present a subgroup analysis by disorder. For pooling, a random-effects model will be used.

**Ethics and dissemination** Ethics approval is not required for this systematic review. The study findings will be submitted to a relevant peer-reviewed journal for dissemination and presented at relevant conferences.

**PROSPERO registration number** CRD42023429996.

## STRENGTHS AND LIMITATIONS OF THIS STUDY

⇒ This study follows Cochrane Handbook for Systematic Review of Interventions guidelines.
⇒ Risk of bias (ROB) will be assessed using the Cochrane Back Review Group's recommended ROB2 tool.
⇒ Strength of the evidence will be evaluated by using the Grading of Recommendations, Assessment, Development and Evaluation (GRADE) pro approach.
⇒ A limitation of the study is that results from relevant studies not published in English, Dutch or German could be missed.

## INTRODUCTION

Upper quadrant musculoskeletal disorders (UQMD), comprising of cranial, cervical, shoulder and upper extremity disorder, are among the most frequently reported disorders in clinical practice.[1 2] They are the second most common cause of work-related musculoskeletal disorders, with only lower back pain being more common.[3]

Most clinical guidelines for cervical and upper limb disorders suggest non-surgical management, as a first treatment option for UQMD.[4–9] Spinal manipulation is one such management approach used in clinical practice. Manipulation is defined as 'a passive, high velocity, low amplitude thrust (HVLAT) applied to a joint complex within its anatomical limit with the intent to restore optimal motion, function, and/or to reduce pain' in the International Federation of Orthopaedic Manipulative Physical Therapists' 'Glossary of Terms'.[10] Thoracic high velocity low amplitude thrust (Tx-HVLAT) manipulation is an intervention used by both orthopaedic manipulative physical therapists, as well as chiropractors and osteopaths.[11 12] In addition to the emerging evidence supporting its use for UQMD, the thoracic spine is also the most commonly manipulated spinal region.[13 14] One of the reasons for this has been linked to the theory of regional interdependence with the thoracic spine being viewed as a silent

contributor to clinical presentations where a pain source lies elsewhere.[15]

Tx-HVLAT is a recommended best practice management option for individuals with neck pain.[5 16–21] Additionally, recent evidence from systematic reviews supports Tx-HVLAT also as an effective treatment option for other upper quadrant disorders such headache,[22] shoulder pain[23 24] and lateral elbow pain.[25] However, no recent systematic reviews have assessed the effectiveness across UQMD.

Therefore, this systematic review aims to synthesise the current evidence on the effectiveness of Tx-HVLAT for patients with UQMD on (1) primary outcomes (eg, pain, disability and perceived effect) and (2) secondary outcomes (eg, performance measures, psychosocial outcomes and adverse events).

## METHODS AND ANALYSIS

### Protocol and registration

This systematic review was registered in the PROSPERO database and this protocol was prepared following Preferred Reporting Items for Systematic review and Meta-Analysis Protocols (PRISMA-P) guidance. The results will be reported in compliance with PRISMA-2020 guidelines.[26]

### Selection criteria

#### Inclusion criteria

Published randomised clinical trials (RCTs) for which full texts are available in English, Dutch or German will be included.

#### Exclusion criteria

Abstracts for which full reports are not available or studies without the outcome of interest will be excluded.

#### Participants

Adult patients (age ≥18 years), with either short-term (less than 3 months), intermediate (3 months to 1 year) or long-term (more than 1 year) UQMD treated in primary care, hospitals, educational or occupational settings will be included. For the purpose of this review, UQMD comprises of issues of pain and/or disability in the head, neck, upper thoracic spine or upper extremity or conditions having been classified or labelled as UQMD.

#### Interventions

Studies using Tx-HVLAT (with or without cavitation) as an intervention for UQMD will be included. The technique could be provided once or multiple times to a single spinal region or various spinal regions during a single session or over multiple sessions. Co-interventions can also be included within the treatment session if these were also included in the comparison group. This allows for differences in treatment effect to be attributed to the addition of Tx-HVLAT in the experimental group.

#### Comparison

Comparisons which will be evaluated shall consist of: (1) placebo, sham manipulation, waiting list control or no treatment or (2) other type(s) of conservative (ie, non-surgical) treatment.

#### Outcome measures

Following guidelines as advised by the Cochrane Back Review Group in establishing our primary outcomes, studies will be included that used at least one of the outcome measures that are considered to be the most important, namely: pain intensity, global perceived effect (eg, proportion of patients recovered, subjective improvement of symptoms), disability (eg, Neck Disability Index, Bournemouth Neck Questionnaire, Shoulder Pain and Disability Index), return to work (eg, days off work) or quality of life (eg, EuroQol 5 Dimension or EQ-5D).

Outcomes of physical examinations (eg, range of motion, spinal flexibility, muscle strength, upper limb nerve tension testing), and psychosocial outcomes (eg, anxiety, depression, pain behaviour) will be considered as secondary outcomes. Other outcomes such as drug consumption or adverse side effects will also be considered as secondary outcomes.

### Search strategy

The search strategy will follow the recommendation by the Cochrane Handbook for Systematic Review of Interventions.[27] The following electronic databases will be searched from inception: the Cochrane Controlled Trials Register, MEDLINE, EMBASE, CINAHL, Index to Chiropractic Literature and the PEDro database. Only studies of which the full text is available in English, German or Dutch language will be included. We will use Medical Subject Headings (MeSH) (MEDLINE), Thesaurus (EMBASE, CINAHL) and free-text words. Combinations will be made based on (1) localisation (eg, head, neck, thoracic spine or upper extremity); (2) disorder (eg, UQMD, headache, neck pain, thoracic spinal pain, shoulder pain, elbow pain or upper extremity disorders); (3) intervention (eg, HVLAT, manipulation, manual therapy, physiotherapy, physical therapy, chiropractic) and (4) design: (randomised clinical trial or randomised controlled trial). Manual searches of published review bibliographies and reference lists of primary studies will be undertaken to search for possible studies not captured by the electronic searches. For the search strategies, please refer to online supplemental appendix 1.

A research librarian together with a review author (ET) will perform the electronic searches. The search results will be uploaded and managed using EndNote V.20 software (Clarivate Analytics, London, UK). Two review authors (ET and GT) will independently screen and select potentially eligible studies. First, the title and abstract will be screened for eligibility. Second, the full-text papers will be assessed to ascertain whether the study meets the inclusion criteria regarding design, participants and interventions. Disagreements on inclusion will be

resolved by discussion or through arbitration by a third review author (MdG).

## Risk of bias assessment

Two review authors (ET and GT) will independently assess the risk of bias (ROB) using the Cochrane Back Review Group's recommended ROB2 tool.[28] This tool is structured into five different domains of bias (arising from the randomisation process, due to deviations from intended interventions, due to missing outcome data, in measurement of the outcome and in selection of the reported result). Within each domain, the assessment comprises: a series of signalling questions, a judgement about ROB for the domain, free-text boxes to justify responses to the signalling questions and risk-of-bias judgements and optional free-text boxes to predict (and explain) the likely direction of bias. In case of a 'sham' manipulation being the comparator, in the ROB assessment V.2.1 this item will be rated as 'perhaps' in cases where the authors do not explicitly show that participants were unaware of their assigned intervention. When disagreement persists, a third review author (MdG) will be consulted. A low ROB is defined as being judged to be at low ROB for all domains for the result.[29]

## Data extraction

Using a standardised data extraction form, two review authors (ET and GT) will independently extract the data (including sample size, participant characteristics, inclusion & exclusion criteria, type of UQMD, types of interventions and comparators, outcome measures, follow-up times and results) of the included RCTs. They will compare extractions in a face-to-face meeting. In cases of uncertainty about the data extracted, a third review author (MdG) will be consulted.

## Data analysis

For continuous data, we will calculate standardised mean differences (SMDs) with 95% CIs. SMD will be used because different measures are frequently used to address the same clinical outcome. Where applicable, the weighted mean difference will be calculated. All data from Visual Analogue Scales or Numerical Rating Scales will be converted to scales ranging from 0 to 100, where necessary. For dichotomous outcomes, relative risks and 95% CI will be calculated. If the published article does not provide enough data, we will contact the original authors in an effort to retrieve additional necessary data, with a reminder being sent after 2 weeks.

Where possible we will present a subgroup analysis by disorder (eg, neck pain, shoulder pain, headache) and by comparator intervention. Prior to pooling, clinical heterogeneity sources such as differences in participant characteristics (eg, age, baseline disease severity, ethnicity, comorbidities), types or timing of outcome measurements and intervention characteristics (eg, dose, frequency of dose, training of interventionists) will be assessed through discussion with the research team.[30] If the research team decides pooling is appropriate, subgrouping and meta-analysis will be considered. For pooling a random-effects model will be used.[31] If multiple time points of outcome are reported, we will report (1) immediate, (2) closest to 6 weeks and (3) closest to 3 months.

RevMan Analyses (RevMan V.5.3) will be used to analyse the data. The inter-observer reliability of the ROB assessments will be calculated using Kappa and categorised agreement as poor (0.0), slight (0.0–0.2), fair (0.21–0.4), moderate (0.41–0.6), substantial (0.61–0.8), or almost perfect (0.81–1.0).[32]

## Strength of the evidence

The overall quality of the evidence will be evaluated using the GRADE method.[33] The quality of the evidence will be based on five principal factors: (1) limitations in study design (downgraded when >25% of the participants are from studies with a high ROB), (2) inconsistency of results (downgraded when there is statistical heterogeneity ($I^2$>40%) or inconsistent findings (defined as ≤75% of the participants reporting findings in the same direction)), (3) indirectness (eg, generalisability of the findings), (4) imprecision (downgraded when the total number of participants across studies is <300 for each outcome) and (5) other considerations, such as reporting bias. The quality of the evidence will be downgraded by one level when one of the factors described above is met.[29 34]

Two independent reviewers (ET and GT) will grade the quality of evidence.

Single studies will be considered inconsistent and imprecise (ie, sparse data) and provide 'low quality evidence', which can be further downgraded to 'very low quality evidence' if there are also limitations in design or indirectness. The following grading of quality of the evidence will be applied[33]:

► High quality evidence: further research is very unlikely to change confidence in the estimate of effect.
► Moderate quality evidence: further research is likely to have an important impact on confidence in estimate of effect and may change the estimate.
► Low quality evidence: further research is very likely to have an important impact on confidence in estimate of effect and is likely to change the estimate.
► Very low quality evidence: very little confidence in the effect estimate.
► No evidence: no RCTs were identified that addressed this outcome.

## Patient and public involvement

The research question in this study forms part of a larger discussion within our patient and public involvement meetings as part of an existing programme of a multi-centre research programme that is focused on UQMD. Patients will not be involved in the data collection or analysis.

## ETHICS AND DISSEMINATION

Ethics approval is not required for this systematic review. The study findings will be submitted to a relevant peer-reviewed journal for dissemination and presented at relevant conferences.

## DISCUSSION

Thoracic spine manipulation (Tx-HVLAT) is reported to be one of the most often used treatment modality in multimodal management strategies, not only for thoracic spinal pain but also for many other UQMD.

Findings from this study may assist clinicians and researchers in formulating an individualised management plan for patients with UQMD. By assessing the effectiveness of Tx-HVLAT for different UQMD, clinicians will be better able to choose when to incorporate Tx-HVLAT as an effective treatment modality in evidence based multimodal management strategies, instead of using a standardised 'one size fits all' approach.

Findings from this study will also serve a need both clinically and within the contemporary literature to inform further research (eg, on efficacy, cost effectiveness). We also aim to compare and contrast this study's findings with previously published systematic reviews.[18–22 24]

While results from relevant studies not published in English or Dutch or German might not be analysed, we feel this will not impact the final outcome as it has been reported that exclusion of trials reported in a language other than English does not significantly affect the results of meta-analyses.[35]

**Contributors** MdG devised the initial focus of this systematic review with the help of all authors. ET and GT are postgraduate research students, MdG is the lead supervisor, NRH and DF are co-researchers. ET drafted the initial protocol manuscript with lead and co-supervisors providing guidance on methodological decisions and proposed analyses. All authors have contributed subject-specific expertise. All authors will contribute to data interpretation, conclusions and dissemination. All authors have read, contributed to and agreed to the final manuscript. DF and MdG are both guarantors of the study.

**Funding** This study will be conducted as part of a postgraduate research project though the University of Birmingham, UK, and as such received no specific grant from any funding agency in the public, commercial or not-for-profit sectors.

**Competing interests** None declared.

**Patient and public involvement** Patients and/or the public were not involved in the design, or conduct, or reporting, or dissemination plans of this research.

**Patient consent for publication** Not applicable.

**Provenance and peer review** Not commissioned; externally peer reviewed.

**ORCID iDs**
Erik Thoomes http://orcid.org/0000-0001-6375-2267
Nicola R Heneghan http://orcid.org/0000-0001-7599-3674
Deborah Falla http://orcid.org/0000-0003-1689-6190

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
