## [Reviewer comments · BMJ Open]

ARTICLE DETAILS

TITLE (PROVISIONAL)	Effectiveness of thoracic spine manipulation for upper quadrant musculoskeletal disorders: protocol for a systematic review
AUTHORS	Thoomes, Erik; Tilborghs, Gus; Heneghan, Nicola; Falla, Deborah; de Graaf, Marloes

VERSION 1 – REVIEW

REVIEWER	Glissmann Nim, Casper University Hospital of Southern Denmark
REVIEW RETURNED	30-Jun-2023

GENERAL COMMENTS	General comments The protocol paper “Effectiveness of thoracic spine manipulation on upper quadrant musculoskeletal disorders; protocol for a systematic review.” The protocol describes an ongoing systematic review to assess thoracic SMT for numerous conditions where regional interdependence could be considered relevant. The review is original in the sense that it is broader than previous reviews and focuses on many different disorders. Overall, the protocol reads well, is transparent, and I enjoyed the length of it. I have some considerations to improve clarity, but also some methodological considerations that I would advise the authors to implement or list as limitations. Introduction I would recommend that you make the introduction more “profession independent.” There are 4 places where specific professions are mentioned, with PT listed across. If you want to mention professions, I will do it in your section “High Thoracic high-velocity low amplitude thrust (Tx-HVLAT) manipulation is an intervention used by both orthopedic manipulative physical therapists as well as chiropractors.” I would list the professions (all or none of them) here. Then omit it elsewhere. This would broaden the readership of the review and reduce any potential claims of being profession-biased. I would recommend changing the word “update” under the aim to “synthesize” or something similar. Update instantly makes me consider that you are updating a review, which you are not, but creating a new one across disorders and outcomes. Method Inclusion criteria: By only including the languages selected, you will likely miss some papers written in Chinese or Spanish; I would strongly recommend adding these to your list or mentioning them as a limitation. Do you intend to exclude grey literature? Participants:
---

	Numerous studies take place in educational settings, e.g., PT or Chiropractic university clinics. Are you not including these? Comparisons: From a not published review, I can state that there are quite large differences in effect sizes of guideline-recommended therapies (e.g., exercise therapy) and non-recommended therapies (e.g., stretching or ultrasound), with non-recommended therapies being closer to the control interventions you use here. The data basis may not be sufficient to split these. Still, I would consider adding this sensitivity analysis to your published review if possible. Outcomes: I would ensure that there is consistency between this section and the aim. So primary outcomes could be pain and disability, and secondary outcomes could be basically anything else. I should be able to obtain this from the aims. Search: Here it seems like you are including a grey little. Please be consistent in the reporting. I would strongly recommend that you search PEDro as well (https://chiromt.biomedcentral.com/articles/10.1186/s12998-022-00468-8) “.” missing and a typo at “Two reviewer authors” Dates are missing, or at least state whether you have a lower limit. If I understand Appendix 1, the search is already complete. Thus, I would add the dates here. Moreover, please be prepared to update the search when you are ready to publish; otherwise, a reviewer will likely ask you anyway. In addition, you list numerous systematic reviews in the introduction and discussion and even state that you intend to update these. Why do you not include the papers previously located in relevant systematic reviews directly from the reviews? Data extraction: Please clarify: “They will compare their forms for consistency, precision and accuracy.” How? And how will you quantify this? Dissemination plan: “To ensure methodological rigor and ensure publication bias, this study protocol will be submitted to an open-access peer-reviewed journal.” I find this statement redundant because the reader will read the study protocol. Alternatively, write it in past tense. Limitations: Please add some items listed above if you keep the protocol the same. (e.g., limited languages and databases). Also, there are no limitations; please reconsider if there is something else you still need to consider. Discussion Please clarify: “We also aim to contrast this study’s findings with previously published systematic reviews” I do not understand the statement. Abstracts Please see my comment on the aim and outcomes. Other comments: Appendix 1 is challenging to interpret.
--	---

REVIEWER	Downie, Aron Macquarie University, Faculty Science and Engineering
REVIEW RETURNED	03-Jul-2023

GENERAL COMMENTS	The research topic is valuable to the field of management of MSK conditions. The protocol is generally well written and follows appropriate reporting guidance (PRISMA-P) with RoB assessment procedures as per Cochrane Back Review Group. Parts of the methodology should include more detail. This is described below. Reporting Where placebo or *sham* is use as a comparator, consideration of the validity of the comparator (per study) should be also be reported to aid interpretation of effect (e.g. alongside forest plot). For sham-SMT to be reported as valid the method should be cited from literature or group allocation assessed for adequate blinding within the study. This will be captured in RoB item 2.1 “Were participants aware of their assigned intervention during the trial?” but is too blunt when considering trials of SMT. e.g. Michener LA, Kardouni JR, Sousa CO, Ely JM. Validation of a sham comparator for thoracic spinal manipulation in patients with shoulder pain. Man Ther. 2015 Feb;20(1):171–5. Exploration of heterogeneity Detail for the statement is needed: “...clinical heterogeneity sources such as differences in participant characteristics (e.g., age, baseline disease severity, ethnicity, comorbidities)...through discussion with the research team” - What is meant by this?– for example, will the team decide if appropriate to pool, additional subgrouping, or meta-regression be considered – this should be pre-specified as per Cochrane https://training.cochrane.org/handbook/current/chapter-10 - Exerpt from Cochrane: “Explore heterogeneity:... Reliable conclusions can only be drawn from analyses that are truly pre-specified before inspecting the studies’ results, and even these conclusions should be interpreted with caution. Explorations of heterogeneity that are devised after heterogeneity is identified can at best lead to the generation of hypotheses. They should be interpreted with even more caution and should generally not be listed among the conclusions of a review. Also, investigations of heterogeneity when there are very few studies are of questionable value.” Random effects-modelling is appropriate as stated. Timing of outcome measures Detail for the statement is needed: “...timing of outcome measurements...” as this may be a significant source of heterogeneity and is clinically important. Consider reporting by BOTH short-term and long-term where available. If multiple time points are reported, consider reporting (i) immediate and (ii) closest to 12 months.
---

VERSION 1 – AUTHOR RESPONSE

Reviewer: 1

Dr. Casper Glissmann Nim, University Hospital of Southern Denmark

Comments to the Author:

General comments

The protocol paper “Effectiveness of thoracic spine manipulation on upper quadrant musculoskeletal disorders; protocol for a systematic review.” The protocol describes an ongoing systematic review to

assess thoracic SMT for numerous conditions where regional interdependence could be considered relevant. The review is original in the sense that it is broader than previous reviews and focuses on many different disorders.

Overall, the protocol reads well, is transparent, and I enjoyed the length of it. I have some considerations to improve clarity, but also some methodological considerations that I would advise the authors to implement or list as limitations.

Introduction

I would recommend that you make the introduction more “profession independent.” There are 4 places where specific professions are mentioned, with PT listed across. If you want to mention professions, I will do it in your section “High Thoracic high-velocity low amplitude thrust (Tx-HVLAT) manipulation is an intervention used by both orthopedic manipulative physical therapists as well as chiropractors.” I would list the professions (all or none of them) here. Then omit it elsewhere. This would broaden the readership of the review and reduce any potential claims of being profession-biased.

Thank you for this recommendation; we have edited the text accordingly

I would recommend changing the word “update” under the aim to “synthesize” or something similar. Update instantly makes me consider that you are updating a review, which you are not, but creating a new one across disorders and outcomes.

Thank you for this recommendation; we have edited the text accordingly

Method

Inclusion criteria:

By only including the languages selected, you will likely miss some papers written in Chinese or Spanish; I would strongly recommend adding these to your list or mentioning them as a limitation. As we are not fluent enough to assess studies written in other languages than mentioned, we have mentioned the consequences of this in our limitations

Do you intend to exclude grey literature?

We do and have mentioned this under “search strategy”

Participants:

Numerous studies take place in educational settings, e.g., PT or Chiropractic university clinics. Are you not including these?

Thank you for this suggestion; we have added educational settings in the text.

Comparisons:

From a not published review, I can state that there are quite large differences in effect sizes of guideline-recommended therapies (e.g., exercise therapy) and non-recommended therapies (e.g., stretching or ultrasound), with non-recommended therapies being closer to the control interventions you use here. The data basis may not be sufficient to split these. Still, I would consider adding this sensitivity analysis to your published review if possible.

By using the term “(b) other types of conservative therapy”, our aim is to include all therapies including those you mention. Therefore we have added the term “non-surgical” for clarification.

Outcomes:

I would ensure that there is consistency between this section and the aim. So primary outcomes could be pain and disability, and secondary outcomes could be basically anything else. I should be able to obtain this from the aims.

Thank you for this, we have aligned the aim in the Introduction with the outcomes in the Methods section.

Search:

Here it seems like you are including a grey little. Please be consistent in the reporting.
Yes, as we mentioned above: we are indeed.

I would strongly recommend that you search PEDro as well
(<https://chiromt.biomedcentral.com/articles/10.1186/s12998-022-00468-8>)

Thank you for this study reference. On the basis of this we have added the PEDro database as well.

“.” missing and a typo at “Two reviewer authors”
We have corrected this.

Dates are missing, or at least state whether you have a lower limit.
We have added these.

If I understand Appendix 1, the search is already complete. Thus, I would add the dates here.
Moreover, please be prepared to update the search when you are ready to publish; otherwise, a reviewer will likely ask you anyway.

We're sorry about the confusion; we uploaded a pilot search document we created with the research librarian/ information specialist giving us number of hits in order to provide us an idea of the level of sensitivity / specificity of the developed search strategy up to that point in time. We have edited it slightly since then, but the final search is not yet executed. We have now uploaded this final search strategy.

In addition, you list numerous systematic reviews in the introduction and discussion and even state that you intend to update these. Why do you not include the papers previously located in relevant systematic reviews directly from the reviews?

We will include these from the list of published reviews as we mention under “search strategy”

Data extraction:

Please clarify: “They will compare their forms for consistency, precision and accuracy.” How? And how will you quantify this?

The two review authors will compare the data they have extracted from each study during a face -to-face meeting and discuss any discrepancies. We have added some more information to clarify this.

Dissemination plan:

“To ensure methodological rigor and ensure publication bias, this study protocol will be submitted to an open-access peer-reviewed journal.” I find this statement redundant because the reader will read the study protocol. Alternatively, write it in past tense.

We have deleted this as indeed it is redundant.

Limitations: Please add some items listed above if you keep the protocol the same. (e.g., limited languages and databases). Also, there are no limitations; please reconsider if there is something else you still need to consider.

We have added the language restriction as a limitation.

Discussion

Please clarify: “We also aim to contrast this study’s findings with previously published systematic reviews” I do not understand the statement.

What we mean by this statement is that we will compare our study findings with the outcome of previous reviews especially those we mention in the Introduction section. We have added some text to clarify this.

Abstracts

Please see my comment on the aim and outcomes.

We have addressed your comments as we mentioned above.

Other comments:

Appendix 1 is challenging to interpret.

We have edited the search strategy and hope it is now less challenging.

Reviewer: 2

Dr. Aron Downie, Macquarie University

Comments to the Author:

bmjopen-2023-076143

Effectiveness of thoracic spine manipulation on upper quadrant musculoskeletal disorders; protocol for a systematic review.

The research topic is valuable to the field of management of MSK conditions. The protocol is generally well written and follows appropriate reporting guidance (PRISMA-P) with RoB assessment procedures as per Cochrane Back Review Group.

Parts of the methodology should include more detail. This is described below.

Reporting

Where placebo or *sham* is used as a comparator, consideration of the validity of the comparator (per study) should also be reported to aid interpretation of effect (e.g. alongside forest plot). For sham-SMT to be reported as valid the method should be cited from literature or group allocation assessed for adequate blinding within the study. This will be captured in RoB item 2.1 "Were participants aware of their assigned intervention during the trial?" but is too blunt when considering trials of SMT.

e.g. Michener LA, Kardouni JR, Sousa CO, Ely JM. Validation of a sham comparator for thoracic spinal manipulation in patients with shoulder pain. *Man Ther.* 2015 Feb;20(1):171–5.

We agree that a valid "sham" HVLT is difficult to execute and we are aware of the mentioned Michener et al., 2015 study. This is why, in the ROB assessment 2.1 we will rate this item as "perhaps" in cases where the authors do not explicitly show that participants were unaware of their assigned intervention. This will then also have consequences for item 4.4 in the ROB especially for self-report items.

Exploration of heterogeneity

Detail for the statement is needed: "...clinical heterogeneity sources such as differences in participant characteristics (e.g., age, baseline disease severity, ethnicity, comorbidities)...through discussion with the research team"

- What is meant by this?— for example, will the team decide if appropriate to pool, additional subgrouping, or meta-regression be considered – this should be pre-specified as per Cochrane <https://training.cochrane.org/handbook/current/chapter-10>

- Excerpt from Cochrane: "Explore heterogeneity:... Reliable conclusions can only be drawn from analyses that are truly pre-specified before inspecting the studies' results, and even these conclusions should be interpreted with caution. Explorations of heterogeneity that are devised after heterogeneity is identified can at best lead to the generation of hypotheses. They should be interpreted with even more caution and should generally not be listed among the conclusions of a

review. Also, investigations of heterogeneity when there are very few studies are of questionable value.”

Random effects-modelling is appropriate as stated.

Thank you for this comment. We are aware of the complexity of assessing clinical heterogeneity. If, after discussion, the research team in consensus feel pooling is appropriate, indeed sub-grouping or meta-analysis will be considered. We have added this information to the text.

Timing of outcome measures

Detail for the statement is needed: “...timing of outcome measurements...” as this may be a significant source of heterogeneity and is clinically important. Consider reporting by BOTH short-term and long-term where available. If multiple time points are reported, consider reporting (i) immediate and (ii) closest to 12 months.

Thank you for this comment; we have added information and edited the protocol accordingly.

VERSION 2 – REVIEW

REVIEWER	Glissmann Nim, Casper University Hospital of Southern Denmark
REVIEW RETURNED	11-Aug-2023

GENERAL COMMENTS	I want to thank the authors for providing a short turn-around in their response. Overall, their response is adequate and the paper should be considered for publication. However, there are still some minor things that should be added.  1. PEDro is not mentioned in the abstract 2. I would recommend that the collapse of all non-surgical approaches is a limitation and should be mentioned under limitation or added as a sensitivity analysis. 3. Reviewer 2 has an excellent point about the validity of sham SMT, and the authors have responded but have not modified the manuscript from what I can tell. I would add that this is extracted, and the ROB assessment is based on this critical information. 4. I unfortunately still find this snippet: "They will compare their forms and for consistency, precision and accuracy (during a face-to-face meeting) and discuss any discrepancies." very difficult to follow. Is not as simple as: Two review authors will extract data and compare extractions in a face-to-face meeting?
---

REVIEWER	Downie, Aron Macquarie University, Faculty Science and Engineering
REVIEW RETURNED	28-Aug-2023

GENERAL COMMENTS	The authors have addressed/defended reviewer comments. I have no substantive further suggestions. Please check revisions again - I found one typo (there may be more) P6 L22 "closets to 3 months)" -> also remove last bracket.
---

VERSION 2 – AUTHOR RESPONSE

Reviewer: 1

Dr. Casper Glissmann Nim, University Hospital of Southern Denmark

Comments to the Author:

I want to thank the authors for providing a short turn-around in their response. Overall, their response is adequate and the paper should be considered for publication. However, there are still some minor things that should be added.

1. PEDro is not mentioned in the abstract

Thank you for noticing this oversight; we have now added it.

2. I would recommend that the collapse of all non-surgical approaches is a limitation and should be mentioned under limitation or added as a sensitivity analysis.

We have added the separate (sensitivity) analysis of non-surgical comparators.

3. Reviewer 2 has an excellent point about the validity of sham SMT, and the authors have responded but have not modified the manuscript from what I can tell. I would add that this is extracted, and the ROB assessment is based on this critical information.

We have now added this to the Risk of Bias section.

4. I unfortunately still find this snippet: "They will compare their

forms and for consistency, precision and accuracy (during a face-to-face meeting)

and discuss any discrepancies." very difficult to follow. Is not as simple as: Two review authors will extract data and compare extractions in a face-to-face meeting?

We have simplified this sentence accordingly.

Reviewer: 2

Dr. Aron Downie, Macquarie University

Comments to the Author:

The authors have addressed/defended reviewer comments. I have no substantive further suggestions.

Please check revisions again - I found one typo (there may be more)

P6 L22

"closets to 3 months)"

-> also remove last bracket.

Thank you for picking up this typo; we have checked the document again.